# Graph-Based Deep Learning Model for Forecasting Chloride Concentration in Urban Streams to Protect Salt-Vulnerable Areas

Victor Oliveira Santos [1], Paulo Alexandre Costa Rocha [1,2,*], Jesse Van Griensven Thé [1,3] and Bahram Gharabaghi [1,*]

1   School of Engineering, University of Guelph, 50 Stone Rd E, Guelph, ON N1G 2W1, Canada; volive04@uoguelph.ca (V.O.S.); jesse.the@weblakes.com (J.V.G.T.)
2   Mechanical Engineering Department, Technology Center, Federal University of Ceará, Fortaleza 60020-181, CE, Brazil
3   Lakes Environmental, 170 Columbia St. W, Waterloo, ON N2L 3L3, Canada
*   Correspondence: pcostaro@uoguelph.ca (P.A.C.R.); bgharaba@uoguelph.ca (B.G.)

**Abstract:** In cold-climate regions, road salt is used as a deicer for winter road maintenance. The applied road salt melts ice and snow on roads and can be washed off through storm sewer systems into nearby urban streams, harming the freshwater ecosystem. Therefore, aiming to develop a precise and accurate model to determine future chloride concentration in the Credit River in Ontario, Canada, the present work makes use of a "Graph Neural Network"–"Sample and Aggregate" (GNN-SAGE). The proposed GNN-SAGE is compared to other models, including a Deep Neural Network-based transformer (DNN-Transformer) and a benchmarking persistence model for a 6 h forecasting horizon. The proposed GNN-SAGE surpassed both the benchmarking persistence model and the DNN-Transformer model, achieving RMSE and $R^2$ values of 51.16 ppb and 0.88, respectively. Additionally, a SHAP analysis provides insight into the variables that influence the model's forecasting, showing the impact of the spatiotemporal neighboring data from the network and the seasonality variables on the model's result. The GNN-SAGE model shows potential for use in the real-time forecasting of water quality in urban streams, aiding in the development of regulatory policies to protect vulnerable freshwater ecosystems in urban areas.

**Keywords:** pollution; Credit River; machine learning; graph neural networks; SHAP analysis

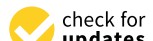



## 1. Introduction

In cold temperate regions of the globe, during winter, deicing substances are often used on roads to improve drivability and road safety, reducing accidents by up to 87% [1–3]. Most commonly, inorganic salts ($NaCl$, $CaCl_2$, $MgCl_2$, $KCl$) are the main tool for ice melting in these regions [3,4]. Their use can be traced back to the United States at the end of the 1930s, and since then, they have been adopted by other countries in the following decades [3,5]. As urbanization has increased, motor vehicles have become more common as a transportation option, leading to an exponential increase in salt usage for road safety improvement [5]. The United States and Canada disperse as much as 24.5 million and 8 million tons of road salt (mainly $NaCl$), respectively [6–9]. While this has a positive impact on road safety during winter, the usage of salt has been shown to have negative consequences, such as the corrosion of automobiles and road infrastructure degradation [9–11].

A plethora of studies have shown that road-applied salt for ice melting is one major anthropogenic sources of increased chloride ($Cl^-$) concentration in soil and water bodies, causing their salinization, more notably so in high urbanized regions, which can reach chloride concentrations as high as 1344 mg/L [2,4,5,12]. The road salt enters the freshwater ecosystems via highway runoff, resulting in high chloride concentrations [4,13]. The high

chloride concentration impairs the freshwater aquatic biota by reducing food availability and decreasing biodiversity [2,14–17].

Many municipalities have installed real-time water quality monitoring stations to accurately assess the environmental impacts of winter road maintenance and reduce road salt application in salt-vulnerable areas. The implementation of low-impact development with the physical modeling of enhanced roadside drainage systems was proposed in previous work as a viable approach to manage chloride concentration for salt-vulnerable areas [18,19]. However, due to the large quantity of data collected at a high frequency, real-time monitoring stations' data are most efficiently analyzed by advanced deep learning models, providing accurate water quality forecasts. When compared to physical-based forecasting models, the data-driven paradigm, such as machine learning (ML) and deep learning (DL) approaches, have been favored by scientists due to their simpler implementation, faster processing times, and inherent capacity to identify complex relationships in the data [20–23].

Different applications of ML models on water quality can be found in the literature. In [24], experimental results pertaining to a Water quality index (WQI) estimation for the Bhavani River, India, showed that the applied artificial neural network (ANN) configuration outperformed their benchmarking models, providing superior accuracy and error values. A similar result was found in another work [25], where ANN was employed for WQI forecasting in Warta River, Poland. Their best-assessed ANN configuration used five hidden neurons in a multilayer perceptron (MLP) structure, returning a root mean square error (RMSE) value of 0.64, proving itself as an essential tool for surface water quality determination. The application of the ML paradigm to groundwater level has been explored in the literature [26]. In this paper, the authors proposed combining wavelet transform (WT) with well-established stand-alone ML models, such as ANN, adaptative neuro-fuzzy inference system (ANFIS), group method of data handling (GMDH), and a least squares support vector machine (LSSVM) to determine groundwater level for the Zarand–Saveh aquifer in Iran for up to 3 months in advance. The results showed that the best model resulted from a combination of WT and LSSVM, achieving RMSE values of 0.05 m and 0.18 m for 1 month and 3 months in advance, outperforming the other assessed models for the same forecasting horizons. In [27], several ML techniques were studied for the application of irrigation water quality for the arid location of the Nfifikh and Cherrate watersheds in Morocco to predict 10 different water quality indexes, including chloride. The results proved that ML can efficiently forecast water quality, with the random forest (RF) model being the most suitable model for chloride prediction in their study.

Future chloride concentration estimates can also benefit from the data-driven paradigm. In [28], the authors propose a data-driven approach to determine future chloride levels in Florida, USA, for groundwater supply. Their approach showed robust performance when forecasting chloride, reaching RMSE and coefficient of determination ($R^2$) values of 28 mg/L and 0.90, respectively. Another data-driven model was implemented by the authors of [21]. Their proposed methodology, real-time chloride forecasting in Grand River, Canada, used an ensemble learning model combining multilayer perceptron MLP and stepwise cluster analysis (SCA). The proposed MLP-SCA achieved good results regarding RMSE (11.58 mg/L) and $R^2$ (0.90). A regression tree-based ML model was suggested by Poor and Ullman [29] for the determination of future levels of nitrate and chloride in the Willamette River, USA. Their analysis increased the $R^2$ values for chloride by 33% when compared to the multiple linear regression model, achieving a final value of 0.75. Their results proved that tree models could handle the complex nonlinearity within the assessed data.

Overall, the data-driven approach shows great potential when applied to the hydrology/environment research area. However, some authors believe there is a deficiency in understanding the application of the deep learning paradigm for predicting chloride levels. To address this problem, the present work proposes to use a cutting-edge approach combining graph theory and DL to assess future chloride concentrations based on spatiotemporal

data for the Credit River located in Ontario, Canada. This work expects to contribute to the field by carrying out the following tasks:

1. Building a state-of-the-art model for chloride concentration, allowing for more accurate and precise results.
2. Conducting an analysis of the contribution of different time lags for the forecasted chloride concentration.
3. Conducting an analysis of the importance of different input variables.

The remainder of this work is structure as follows: In Section 2, the methodology used is presented, followed by Section 3, where the achieved results are shown. In Section 4, there is a discussion of the results, and Section 5 closes the work and contains our conclusions.

## 2. Materials and Methods

### 2.1. Credit River: Characteristics and Dataset

The Credit River is located in Southern Ontario, Canada, just west of Toronto. Its source is located in Orangeville, and the rivers flows until reaching Lake Ontario in a 90 km trajectory [30,31]. The Credit River has a total drainage area of 93,000 ha, and its land composition is split as follows: 35% for agriculture, 27% for urban use, and the remaining 38% comprises natural habitats. The area has an estimated population of 1 million people [32–34]. The river is important for environmental conservation due to its rich aquatic biodiversity and its role as a vital water source for the local population [30,35]. A map showcasing the location of the Credit River watershed and its tributaries is presented in Figure 1, where the red mark represents the reference station, i.e., where the chloride concentration is being forecasted, namely "Credit River @ MGCC", while the green marks show the location of the neighboring stations that provide spatiotemporal data.

However, the highly urbanized Credit River watershed environment adds elevated pollutant concentrations to the river, risking human and animal lives. The present work proposes the use of a graph-based model called GNN-SAGE to estimate future pollutant concentrations (mainly chloride) in the Credit River. Unlike traditional ML and DL paradigms, graph-based models such as GNN-SAGE can process multi-spatiotemporal data, satisfactorily identifying the underlying relationship between the input variables and the target variable when used in forecasting applications [36,37]. Its ability to extract spatiotemporal information from data is critical in the current study due to chloride concentration relying on both temporal and spatial features.

The dataset used for this study contains historical data concerning the Credit River from 2016 to 2020. The historical data have a time resolution of 15 min. With the exception of precipitation (which was summed), all the variables were resampled to 1 h intervals to calculate the average value for each one of them before being fed to the assessed models. The stations distributed along the river's course measure values for the water's physical-chemical characteristics and weather attributes. Figure 2 shows the correlation between each attribute in matrix form.

Figure 2 depicts the correlation matrix, with darker blue colors indicating highly correlated attributes, and darker red showing a high negative correlation. Figure 2 shows a strong positive correlation between chloride levels and water conductivity. Against common sense, this may indicate a collinear relationship between these two attributes, meaning that the conductivity information may be already provided to the model by the chloride data, which can hamper the model's performance by increasing its variance [38,39]. Air and water temperatures have a moderately negative correlation with chloride, while dissolved oxygen has a positive correlation. Although the remaining attributes have a weak correlation, these variables may add important information to the model due to the movement of salt dissolved in the river, helping its modeling and, consequently, future chloride concentrations, as suggested by the SHAP analysis presented in this work.

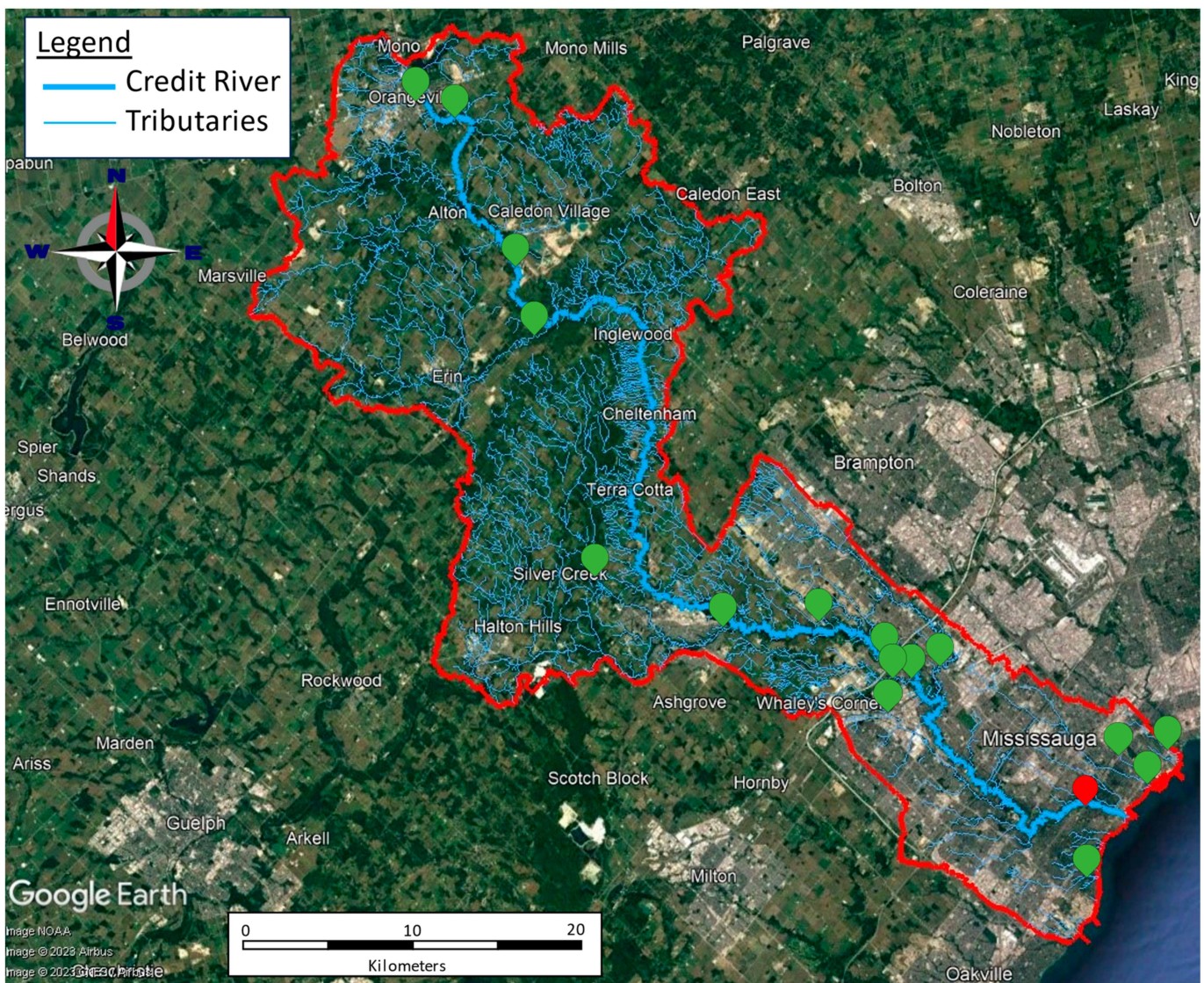

**Figure 1.** Credit River watershed map. The red mark shows the location of the reference station, and the green marks are the neighboring stations.

The dataset was split into two stages: training and validation. The training stage was performed using data from 2016 to 2019, and the validation stage was conducted using data from 2020, as shown in Figure 3. In the figure, the blank spaces represent gaps in the historical data, which were not used in the training phase.

## 2.2. Benchmarking Model

The performances of the Deep Neural Network-based transformer (DNN-Transformer) and the Graph Neural Network Sample and Aggregate (GNN-SAGE) paradigms were evaluated using the benchmarking persistence model. Persistence is a simple forecasting model used as a minimal benchmarking tool. It states that the following attribute measurement is the same as the latest [40,41]. This approach can achieve good results for short forecasting horizons. However, its performance deteriorates for further future horizons as the model cannot track the influence of the dynamics of external factors impacting future values [42,43].

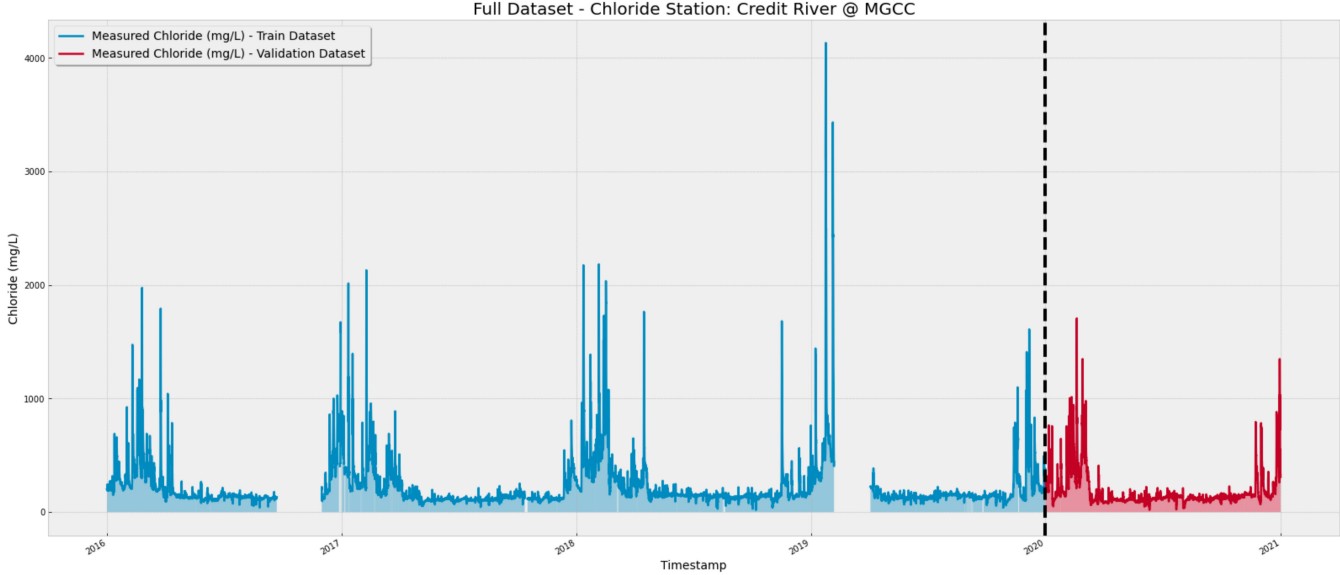

**Figure 2.** Correlation matrix for the measure attributes at Credit River.

**Figure 3.** Dataset split for the training and validation stages. The gaps in the image represent the lack of data for the period.

### 2.3. DNN-Transformer

In the original Transformer structure [44], the encoder embeds data to a context vector using positional encoding and stacks a multi-head attention mechanism, determining how the provided input attends to each other. The encoder output is then fed to the decoder, which generates the most probable forthcoming word for NLP applications [44]. Transformer-based models have exhibited superior or comparative performance during comparisons with recurrent neural networks when applied to different areas, such as speech recognition [45], computer vision [46,47], and time series forecasting [48].

The present study adapted the transformer structure to the proposed regression problem of forecasting chloride concentration using the PyTorch and Scikit-Learn libraries for Python [49,50] (the library's documentation can be accessed at https://pytorch.org/ and https://scikit-learn.org/stable/, accessed on 28 August 2023). The applied architecture for the transformer encoder and the DNN-Transformer architecture are presented in Figures 4 and 5, respectively.

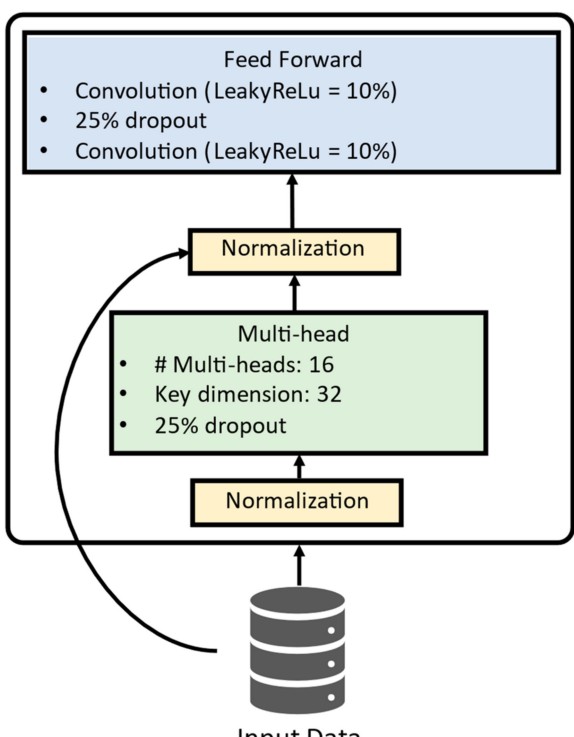

**Figure 4.** Transformer encoder architecture.

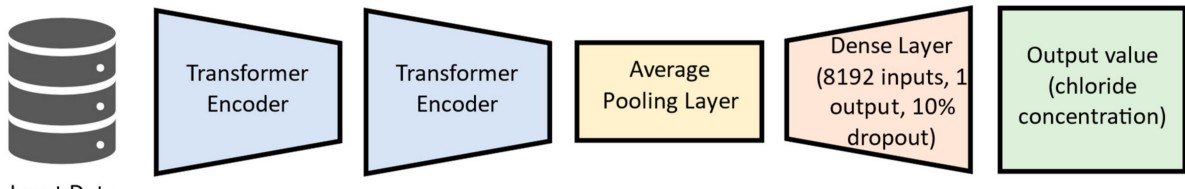

**Figure 5.** DNN-Transformer architecture.

Figure 4 shows that the transformer uses just the encoder structure for the present study. There, the input data are normalized before being fed to the multi-head structure, which is composed of 16 attention-heads, each one with a key dimension equal to 32. After that, the processed data are again normalized with residual information from the original input dataset and then passed to the feed-forward structure, composed of convolutional

layers activated by Leaky ReLu [51], using a 25% dropout. The DNN-Transformer structure, depicted in Figure 5, is composed of two encoders followed by an average pooling layer. A dense layer with 8192 neurons follows the model, which finally outputs the predicted chloride concentration.

### 2.4. GNN-SAGE

The GNN-SAGE was first proposed [52] as a general inductive framework for handling large graph structures. In this approach, nodes are equally sampled around an area of interest during the sampling phase. Afterward, the spatiotemporal information retrieved from these nodes is aggregated by an aggregate operator [53]. This generates an embedding vector representing the node of interest that is also able to generalize unknown data, disregarding the graph's topology and structure [41,52,54]. The GNN-SAGE model's structure enables it to capture complex spatiotemporal patterns between a node and its neighbors, enhancing its forecasting performance compared to traditional ML and DL methods. This results in cutting-edge outcomes when applied to various time series problems [42,43]. The GNN-SAGE model was implemented using Pytorch and Scikit-Learn for Python, and its structure in the context of this study is presented in Figure 6.

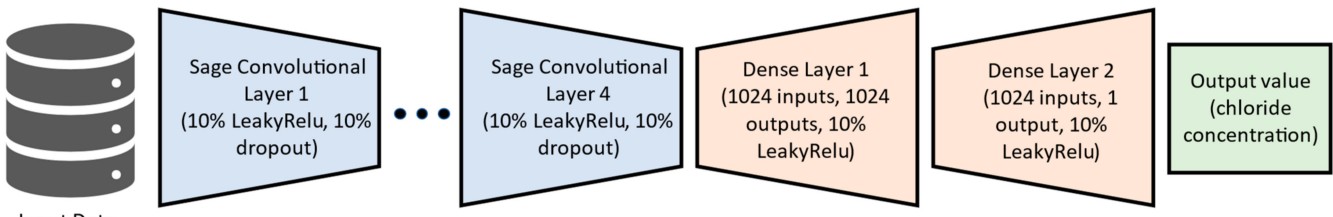

**Figure 6.** GNN-SAGE architecture.

As shown in Figure 6, the spatiotemporal data are fed to the first SAGE convolutional layer using 10% Leaky ReLu as the activation layer and a 10% dropout rate. The convolution process is repeated four times, identifying and extracting relevant structure patterns in the data. After that, the processed data are passed to a sequence of two dense layers using 10% Leaky ReLu, where the forecasted chloride concentration is finally output by the model.

### 2.5. SHAP Analysis

Shapley Additive Explanations (SHAP) is a way to provide insight into how ML models work [55]. The SHAP analysis, based on game theory, calculated the contribution of each input parameter used by the model for forecasting and was implemented into the present work after the model had been trained using the SHAP library (the SHAP library documentation can be read at https://shap.readthedocs.io/en/latest/, accessed on 28 August 2023) by evaluating the model for each situation where one of the independent variables is not used. This way, SHAP can identify relationships among the input data, identifying their influence, importance, and correlation over the model's output [41–43,55]. The determination of the influence of each variable provides deeper insight into how the model provides its results, being a viable option to explain the analyzed ML paradigm locally. The employment of SHAP analysis by those with expertise in different knowledge areas, such as pharmaceutical [56], engineering [57], and social sciences [58], renders it a valuable tool for researchers. In Figure 7, we present a flow chart outlining the tasks performed during our study.

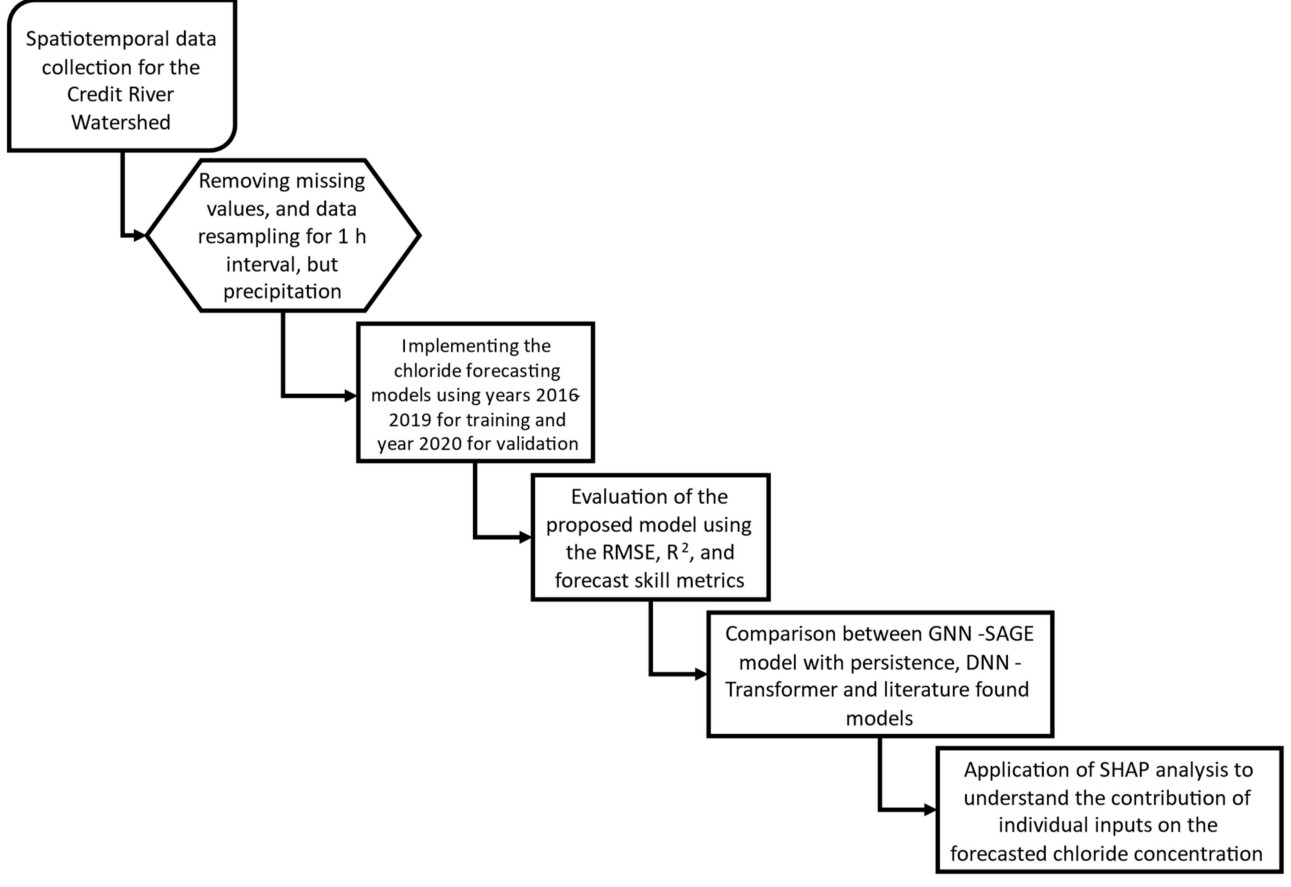

**Figure 7.** Flow chart of the present study.

### 3. Results

*3.1. Size of Time Window Effect*

Figure 8 presents the results for the effect of different time window sizes, i.e., the number of time lags applied as inputs on both the GNN-SAGE and DNN-Transformer models. For this analysis, it was decided that only chloride information would be used.

Figure 8 shows the RMSE for different numbers of time lags for the proposed DNN-Transformer and GNN-SAGE models. Increasing the time window size proved to be beneficial for the graph-based model up to 12 h. Beyond that threshold, the results started to deteriorate. For the DNN-Transformer, incorporating past information enhanced the model's performance for up to 18 h, where more time lag values started to harm the model's performance.

The GNN-SAGE and DNN-Transformer models outperformed persistence, yielding improvements of 12.4% and 11.7%. For the graph-based approach, the best result was obtained using 12 h of past data, reaching an RMSE value of 59.73 ppm, while the DNN-Transformer needed 18 h to provide its best outcomes for an RMSE of 60.24 ppm. Compared to the DNN-Transformer, the proposed GNN-SAGE improved its forecasting by 0.8% for that situation. Based on the results presented in Figure 8, we used 12 time lags for all input variables, including chloride concentrations, to determine future $Cl^-$ values.

*3.2. Chloride Concentration for 6 h Ahead Forecasting Horizon*

The impact of the input variables used on chloride forecasting was evaluated through a step-by-step analysis for a 6 h ahead forecast horizon. At first, in order to forecast future chloride concentrations, the model's sole input was past chloride concentrations. After each test, more input variables were introduced into the model. If the inclusion of a variable improved the model's performance, it was kept as an input; otherwise, it was

discarded. This procedure was repeated until all the input variables described in Figure 2 were assessed, resulting in a selection of variables that only returned the best forecasting values in terms of RMSE. The results for this test are shown in Figure 9, where the lighter the color, the better the error achieved by the model.

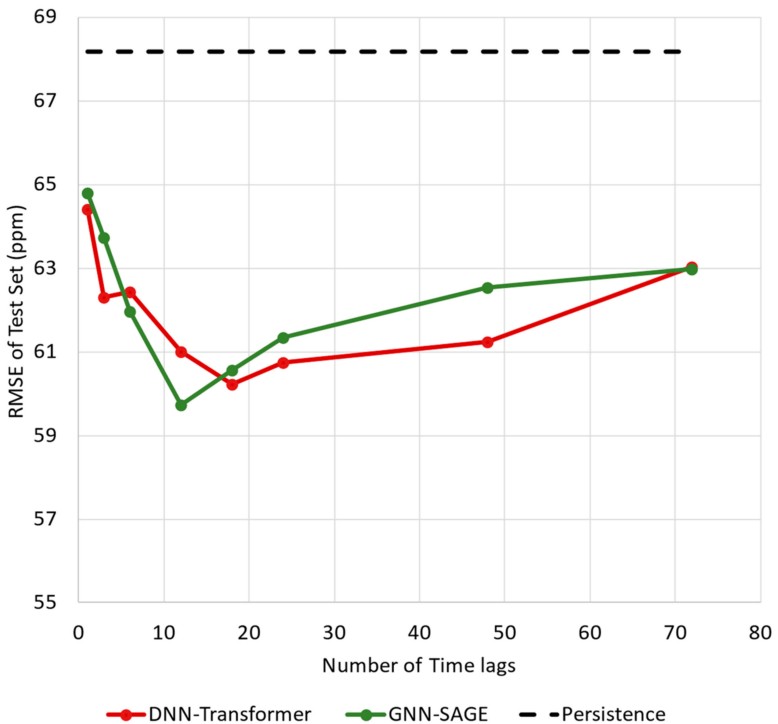

**Figure 8.** Influence of different numbers of time lags in the models' performance.

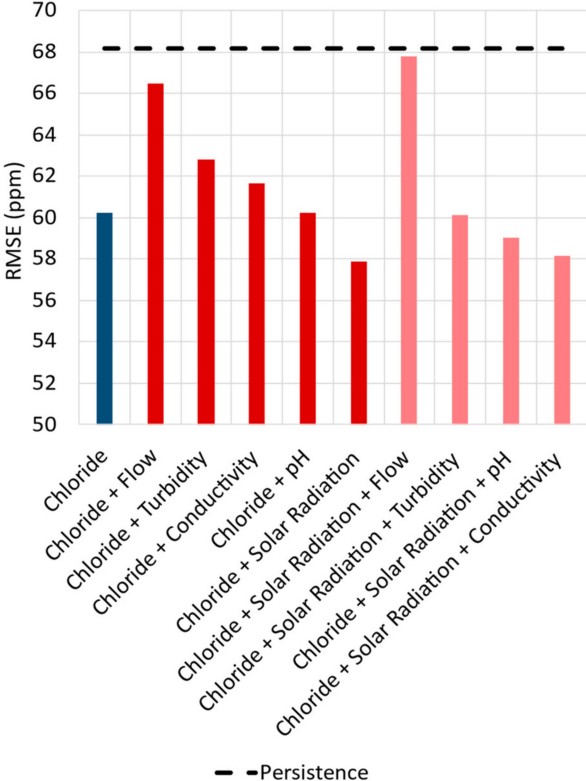

**Figure 9.** The effect of different input variables on the DNN-Transformer performance.

In Figure 9, the best result was achieved when combining chloride and solar radiation, resulting in an RMSE of 57.86 ppm. Adding more than two variables showed no improvement over the transformer performance, indicating that the model could not extract the spatiotemporal information from the additional inputs. Solar radiation, on the other hand, appears capable of providing temporal information in terms of seasonality, both yearly and daily, improving the model's forecasting capacity. The results for forecasting chloride concentrations 6 h in advance are presented in Figure 10.

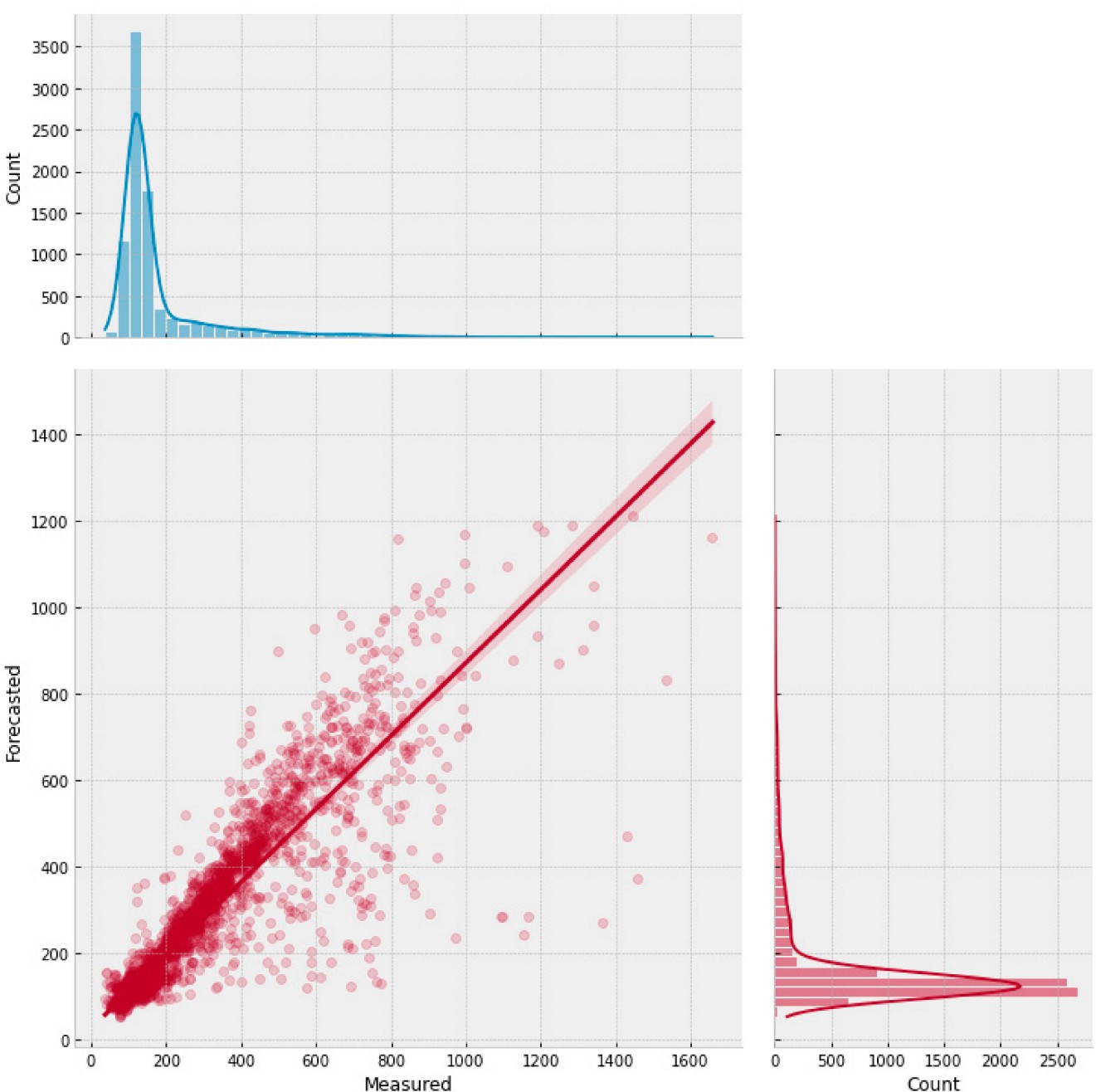

**Figure 10.** Scatter plot with the forecasted and measured chloride concentrations for 6 h in advance (obtained using the DNN-Transformer).

Figure 10 presents a scatter plot for the transformer model and the marginal distributions for the actual and forecasted concentration values. The graph shows good agreement between the measured and actual chloride values, as evidenced the clustered points around the regression line, which reached a coefficient of determination of 82%,

and by both marginal distributions having similar distributions. The DNN-Transformer model reached an RMSE value of 57.86 ppm and an MBE of −1.97 ppm, suggesting a slight underestimation of the forecasted values.

The results regarding the variable testing for the proposed GNN-SAGE model are presented in Figure 11.

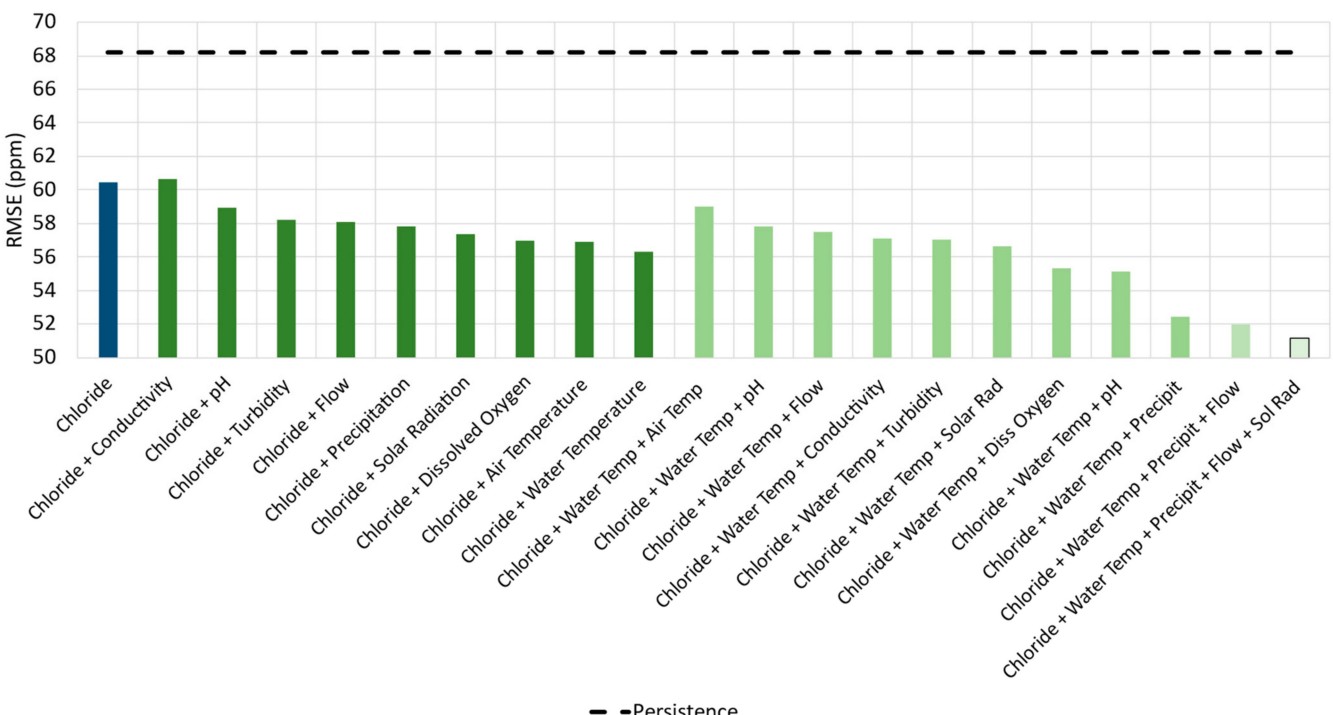

**Figure 11.** The effect of different input variables on the GNN-SAGE performance.

As shown in Figure 11, the model's forecasting ability improved with the inclusion of additional variables. For this case, the best solution was reached using chloride, water temperature, precipitation, flow, and solar radiation, resulting in an RMSE of 51.16 ppm.

The decrease in the DNN-Transformer model's performance can be explained by the fact that it could not identify and extract the spatiotemporal information underlying the dataset. This ultimately prevented the model's generalization of the problem, returning inferior results than the GNN-SAGE. The proposed model, however, could extract and identify the spatiotemporal relationship between input and output variables, improving its generalization and, consequently, its forecasting due to its better understanding of the graph-structured data [37], as verified in previous studies [41–43]. Figure 12 shows a scatter plot for the GNN-SAGE model.

Figure 12 demonstrates that the forecasted and actual data are in good agreement once more. Compared to the deep learning transformer approach, the proposed SAGE model could cluster the points even closer to the regression line, with a more similar marginal distribution of its data and an improved coefficient of determination of 88%. The graph-based paradigm had RMSE and MBE errors of 51.16 ppm and −0.64 ppm, respectively. Compared to the RMSE errors of persistence and the DNN-Transformer, the GNN-SAGE model increased forecasting by 25% and 12%, respectively. These findings show that the GNN-SAGE model can produce more accurate and precise results than the benchmarking models. The superior results for the graph model can be seen in Figures 13 and 14.

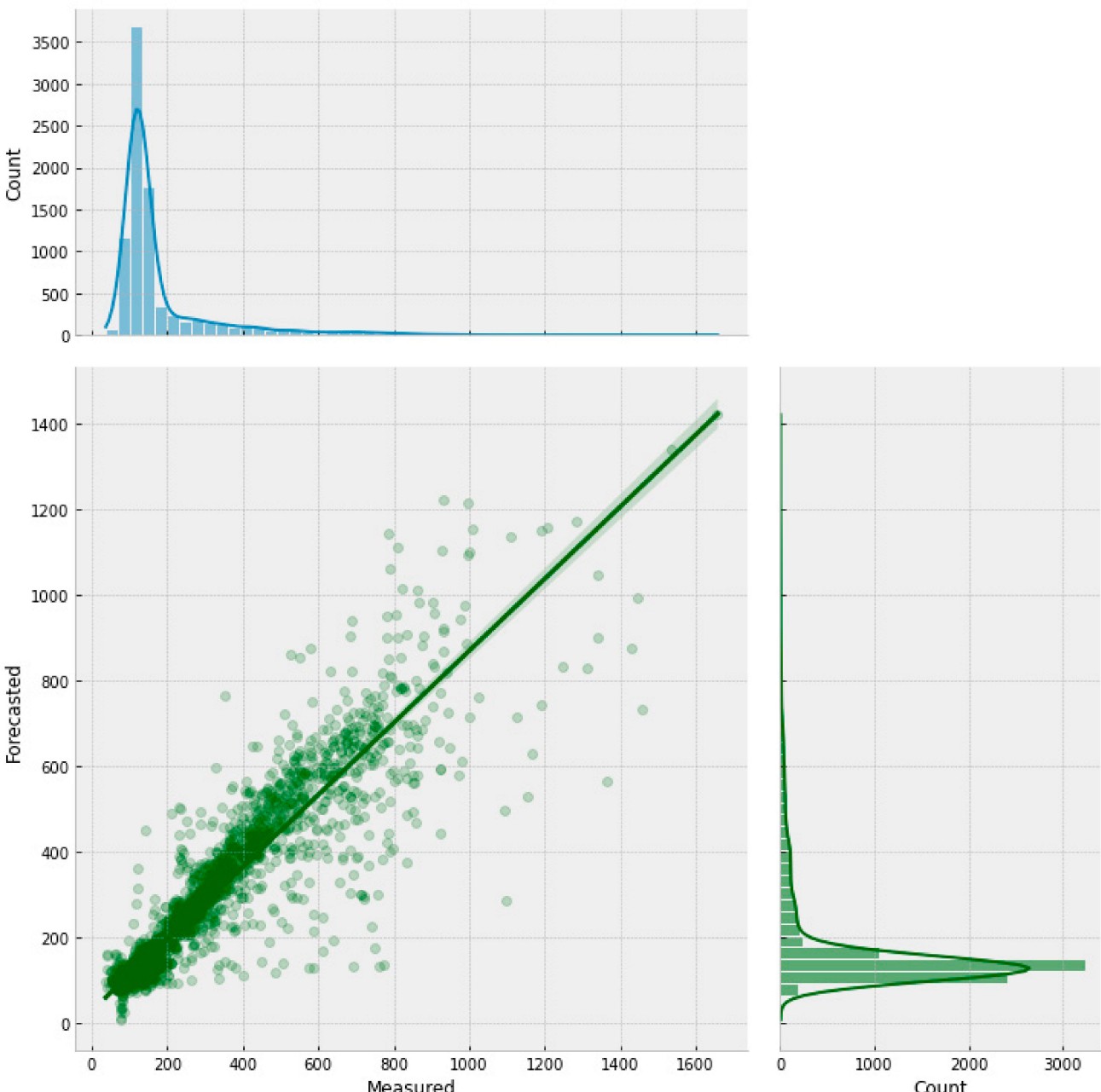

**Figure 12.** Scatter plot with the forecasted and measured chloride concentrations for 6 h in advance (both obtained using GNN-SAGE).

The continuous line in Figure 13 represents the observed chloride values, while the dashed line represents the predicted values. It is possible to visualize that both the DNN-Transformer and the proposed SAGE models can adequately identify the peaks during the assessed period. However, the GNN-SAGE provides more accurate results. While analyzing the period between 15 February 2020 and 1 March 2020, GNN-SAGE closely followed the concentration peak, providing results near to the actual observed chloride concentration values and surpassing the transformer's performance for the same period. Figure 14 presents a closer look at the assessed period.

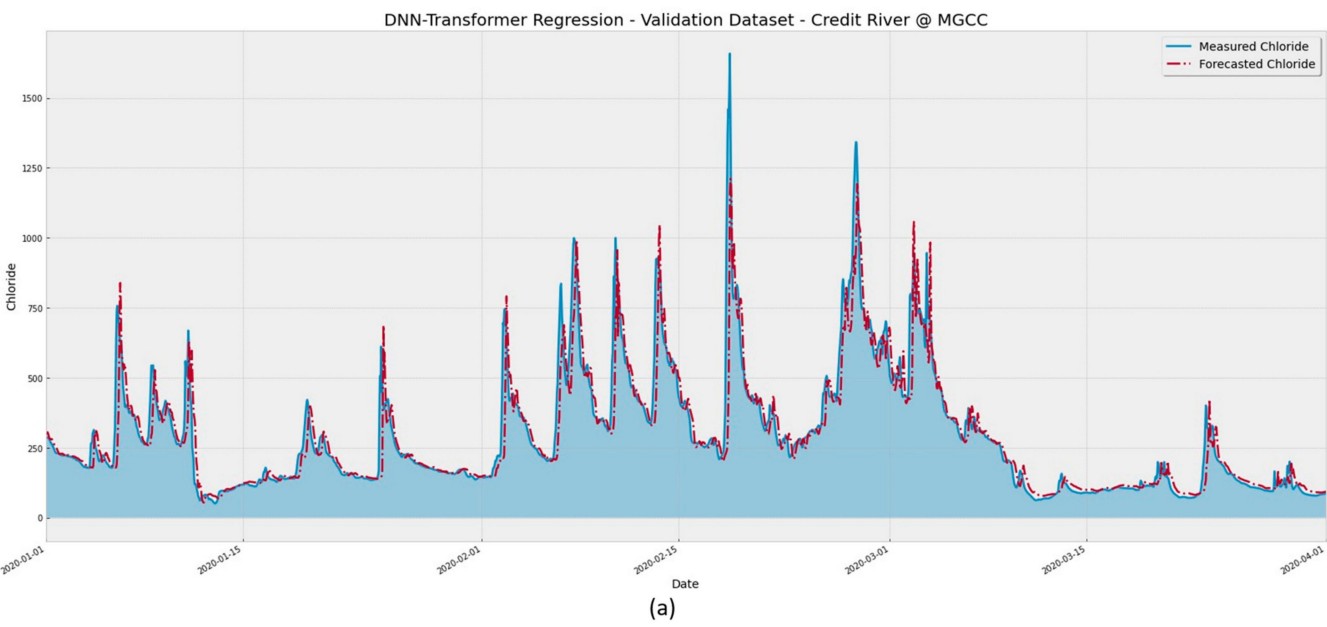

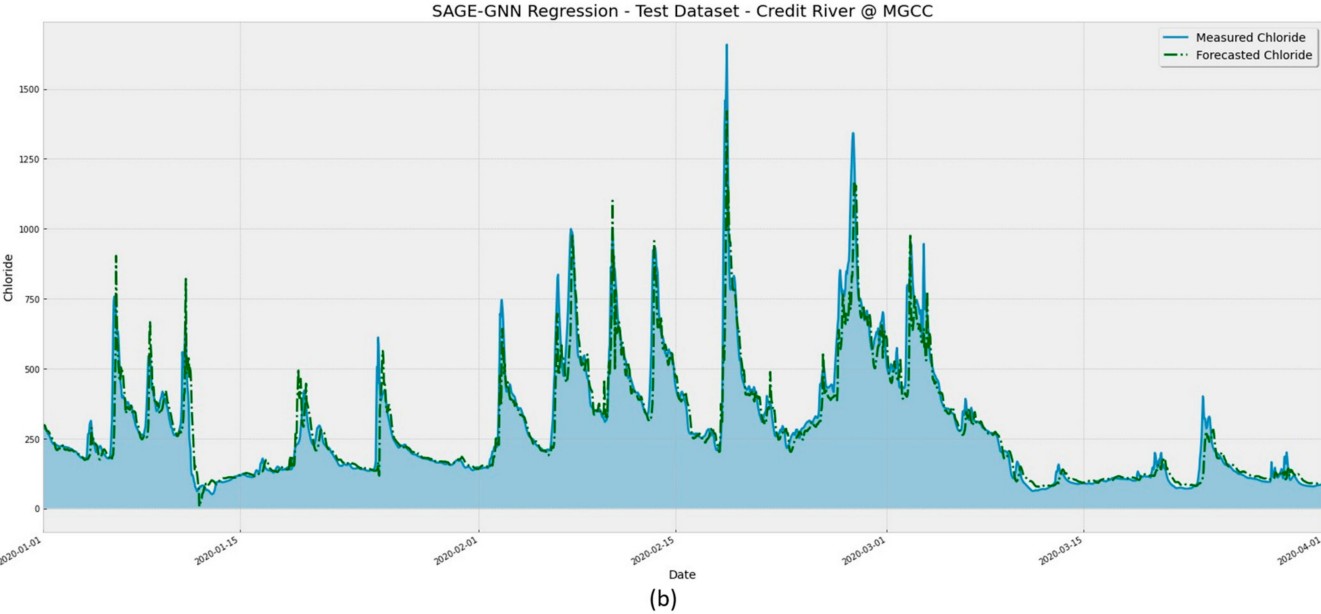

**Figure 13.** Comparison between forecasted and real values for (**a**) DNN-TRANSFORMER and (**b**) GNN-SAGE models for the whole validation dataset.

Figure 14 shows model performance forecasting the chloride concentration at a range narrower than in Figure 13. From this, it is easier to verify that the SAGE model was better at identifying the concentration peaks, as seen for the dates of the 11 and 13 of February 2020. The proposed model reduced the lag between the actual and forecasted values. This lag is known in the literature and can be attributed to the need for more spatiotemporal data for extended time windows as the leading forecasting time increases [59–61]. However, the proposed model reduced this gap when applied to different time series forecasting problems, providing more accurate and reliable predictions in longer-horizon forecasts [42,43].

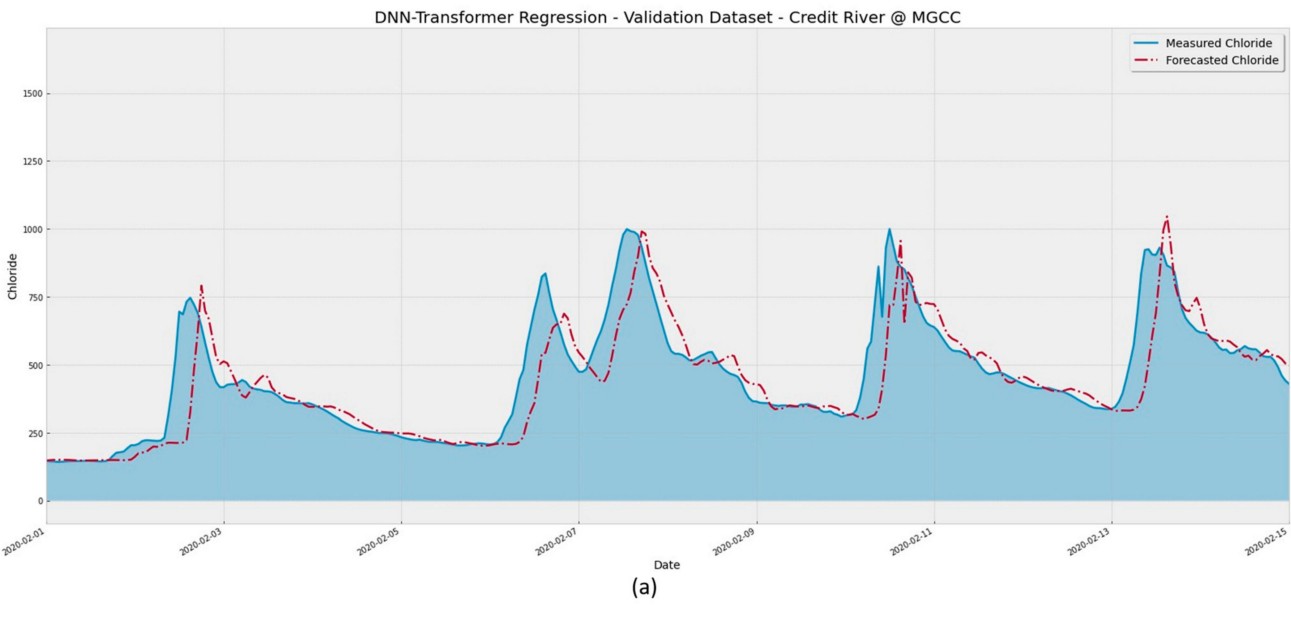

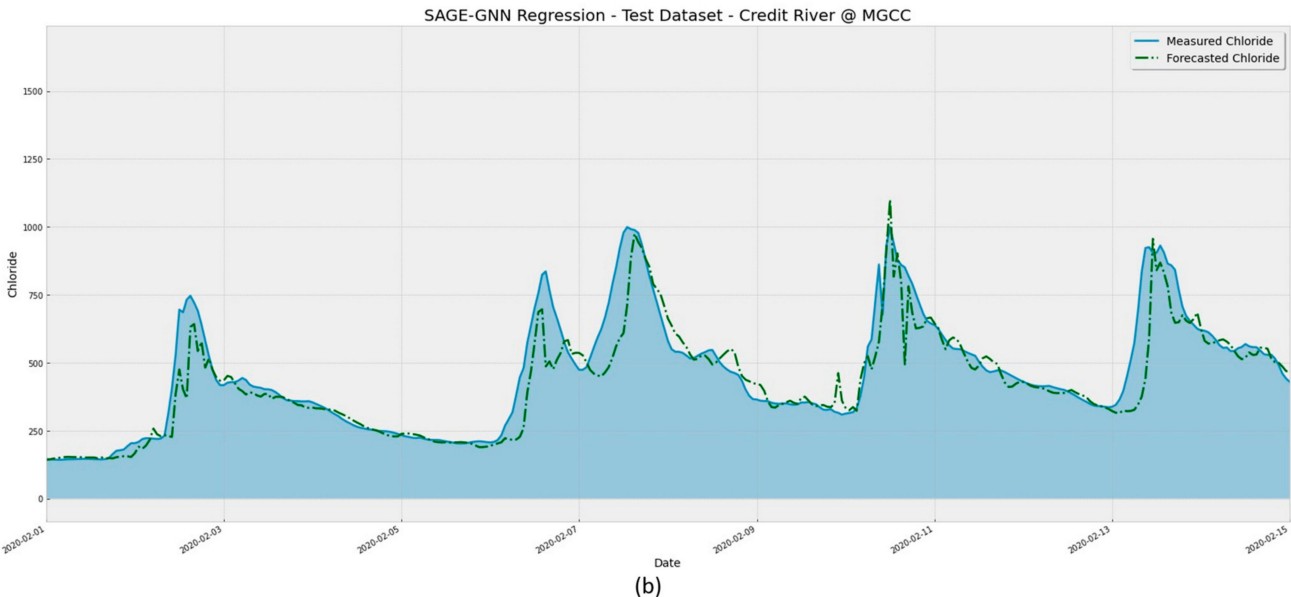

**Figure 14.** Same as Figure 13 for a narrower range, comprising the period from 1 February 2020 to 15 February 2020 for the (**a**) DNN-Transformer and (**b**) GNN-SAGE models.

### 3.3. SHAP Analysis Results

The results of the SHAP analysis are presented in Figure 15. The results are organized in descending order, where the closer to the top, the more important the attribute is for the forecasted value. The rightmost bar in the figure represents the correlation between the variable and the output value. A higher correlation indicates a higher feature value. Furthermore, negative SHAP values indicate that the attribute had a negative influence over the forecasting and vice versa.

In Figure 15, it is possible to state that data from the reference station "Credit River @ MGCC" make a major contribution to the determination of the chloride concentration. Figure 15 also shows that the top three most influential variables for the model's forecasting are the chloride concentrations from the reference station. Moreover, the SHAP results show that neighboring stations contribute to the model's output. These stations provided important information regarding water temperature, which is the fourth most influential attribute; solar radiation, which may provide seasonality information; and flow. This

states the importance of spatiotemporal information coming from the surroundings of the reference station in improving the model's forecasting [41–43,62].

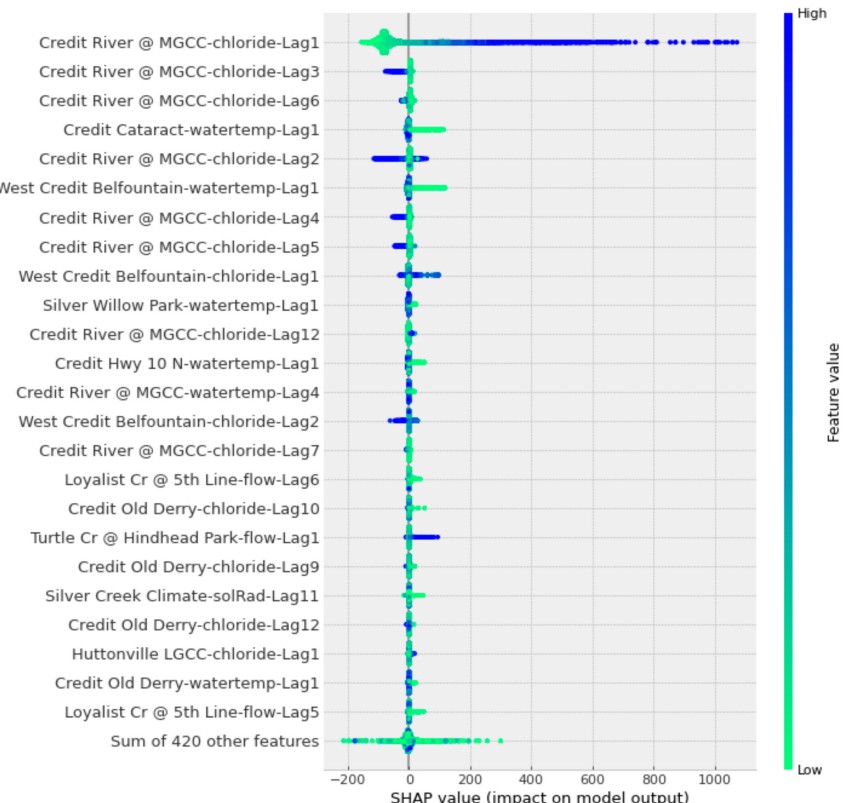

**Figure 15.** SHAP analysis results for forecasting using the GNN-SAGE model.

## 4. Discussion

The GNN-SAGE model proposed in this research has proven to be a reliable tool for estimating future chloride concentrations in the Credit River. Based on graph theory and deep learning, its structure can satisfactorily identify and extract complex spatiotemporal dependencies in data collected from the neighboring stations. Its superior performance for time series applications has been well documented in the literature, where this approach consistently produces state-of-the-art results with respect to forecasting [42,43]. The same behavior was observed in the present work, where the GNN-SAGE paradigm outperformed both persistence and DNN-Transformer models, achieving RMSE and $R^2$ values of 51.16 ppb and 0.88, a substantial improvement over the other assessed models.

The GNN-SAGE approach also reduced the lag between the forecasted and actual measured chloride concentrations, a common phenomenon in time series forecasting as larger forecasting windows require more data [59–61]. The narrowing of this prediction gap is fundamental for providing more accurate and precise future chloride concentrations, allowing for better decision-making and policy development by stakeholders and environmental agencies.

The SHAP analysis we conducted provided an insightful examination of the GNN-SAGE model. Its results showed that the forecasting of $Cl^-$ depends on local and neighboring concentration ion levels. The SHAP analysis also depicts water temperature and solar radiation as other essential variables in chloride forecasting, showcasing the seasonal behavior of chloride.

We compared the results of chloride forecasting found in the literature with those obtained using GNN-SAGE. However, it is important to note that directly comparing different predictive models can be challenging. Each study has its own methodology and unique characteristics, making it difficult to draw direct comparisons between the models.

Furthermore, since WQI forecasting using machine learning has been much more frequently explored than direct chloride prediction, not as many works are available for comparison using this approach. Table 1 compiles metric values for the proposed GNN-SAGE model, and the results found in the literature are presented in Table 2.

**Table 1.** Summary of performance metrics for forecasting using the GNN-SAGE model for a time window that is 6 h ahead.

| Metric | 6 h Ahead |
|---|---|
| RMSE (m) | 51.16 ppb |
| $R^2$ | 0.88 |
| MBE | −0.64 ppb |
| Forecast Skill | 0.24 |

**Table 2.** Literature values for chloride prediction.

| Model | Metric Value | Author |
|---|---|---|
| SCA-MLP | RMSE ($R^2$) 11.58 mg/L (0.90) for 1 h forecasting horizon | Zhang et al. [21] |
| FOS | RMSE ($R^2$) 28.00 mg/L (0.90) | El-Jaat et al. [28] |
| Regression tree | $R^2$ 0.85 | Poor and Ullman [29] |
| Multiple regression analysis | $R^2$ 0.64 | Poor et al. [63] |
| Integrated catchment for $Cl^-$ simulation (INCA-Cl) | $R^2$ 0.45 average for monthly simulated $Cl^-$ concentration | Jin et al. [64] |

In [21], chloride concentration was estimated for a 1 h forecasting horizon for the Grand River, Ontario, Canada. The authors of this study proposed an ML model combining multiple-layer perceptron with stepwise cluster analysis for this task. Their approach is based on ensemble learning, which has been proven to boost time series forecasting results [22]. When comparing the GNN-SAGE model to their results, it is evident that the graph-based approach delivers better error values for a longer horizon (51.16 ppb) but has a slightly lower $R^2$ value of 0.88. This indicates that GNN-SAGE can provide more accurate results than traditional approaches such as SCA-MLP. Another ML learning model is proposed in [29], where the authors employed the regression tree paradigm to estimate future chloride concentrations in the Willamette River watershed in the USA. This work is an improvement over the author's previous study [63], where they first proposed chloride forecasting using multiple regression analysis. Comparing both studies, the tree-based model outperformed the former approach, increasing $R^2$ from 0.64 to 0.85. The proposed GNN-SAGE architecture surpasses the results found in both studies, providing a superior coefficient of determination of 0.88, meaning an improvement of 3.5% over the regression tree forecasting was achieved. In [28], a data-driven approach was used to evaluate future chloride concentrations in Deltona, Florida. Unlike the other studies mentioned, the authors proposed estimating $Cl^-$ concentration for groundwater supply. The used model resulted in an RMSE of 28.00 mg/L and $R^2$ of 0.90. Similarly to the results found in [21], GNN-SAGE once again reduced the RMSE error for the chloride estimation, also exhibiting a slightly reduced coefficient of determination. Again, this suggests the superior performance of the GNN-SAGE over this data-driven approach for chloride forecasting.

The Integrated catchment for $Cl^-$ simulation (INCA-Cl) is a physical-based model for determining future chloride concentrations with daily temporal resolution. It is a dynamic, mass-balance approach that aims to verify the temporal changes in the river's flow path [65]. In [64], INCA-Cl was also used to predict chloride concentrations in Ethiopia. In their work, the physical model reached an average $R^2$ value of 0.45 for monthly $Cl^-$

concentration, indicating inferior performance when compared to the GNN-SAGE result of 0.88. Additionally, it is important to note that one of the main advantages of using ML and DL paradigms over physical-based approaches is its simpler implementation. The INCA-Cl model, for example, needs simulated data for hydrological and soil modeling, as well as a geographical information system file to delineate the watershed sub-catchments before running the model [64–66]. On the other hand, the GNN-SAGE model only requires the measured parameters, as depicted in Figure 2. Also, the INCA-Cl model is not suitable for real-time chloride monitoring due to its daily temporal resolution. In contrast, the GNN-SAGE model can perform intra-hour forecasting, providing instantaneous and punctual values, which is more sophisticated than globally integrated values over time.

By using spatiotemporal information, the GNN-SAGE enhances the concept of temporal auto-regression to a spatiotemporal paradigm, using the measuring station data that have an effect over the target parameter (chloride). Our results show that the GNN-SAGE model is able to properly forecast extreme events for chloride concentration for the assessed forecasting horizon of 6 h. When compared with previous works found in the literature, the proposed GNN-SAGE model offers superior performance for determining future chloride values. Considering both RMSE and $R^2$ metrics, GNN-SAGE was able to overcome traditional ML applications such as MLP, regression trees, and data-driven FOS. For the physical-based INCA-Cl, GNN-SAGE offers a simpler implementing approach and is able to provide intra-hour predictions for chloride concentrations with real-time chloride monitoring applications. Overall, the GNN-SAGE model has been shown to be a superior approach for chloride forecasting.

## 5. Conclusions

Our developed GNN-SAGE model was used to predict chloride concentrations. Our GNN-SAGE model was trained with historical data from 2016 to 2020 collected from stations distributed along the Credit River course. The model was subsequently tested using different data inputs. The best configuration for the proposed graph model was reached using past chloride concentration data, water temperature data, precipitation data, flow data, and solar radiation data, together with a time lag of 12 h, as input variables.

To assess the proposed model, the DNN-Transformer and the benchmarking persistence models were also evaluated for chloride forecasting. When compared to the other two models for a 6 h forecasting horizon, the GNN-SAGE model outperformed expectation both in terms of RMSE and $R^2$ evaluation metrics, achieving values of 51.16 ppb and 0.88, respectively. A SHAP analysis was also conducted for this study to gain better insight into the model's forecasting. The results of our SHAP analysis provided an understanding of how spatiotemporal data from neighboring stations majorly affect the GNN-SAGE results. The SHAP analysis indicated that seasonality plays an important part in chloride estimation and that flow, water temperature, and solar radiation are also relevant attributes.

A comparison of the GNN-SAGE model with results from the literature revealed that it delivers state-of-the-art performance for estimating chloride levels, achieving superior RMSE values and comparable $R^2$ values. This comparison deems the proposed GNN-SAGE as a reliable tool for chloride forecasting, providing accurate and precise estimations of $Cl^-$ up to 6 h in advance.

Future works may address some of the model's limitations, such as its geographical limitations. To overcome this hindrance, the model can be trained and validated in different rivers with different watershed sizes and hydrological structures. Also, the model's performance can be verified on further forecasting horizons, which would give further insight into the model's functioning and allow for different decision-making strategies regarding chloride pollution.

The accurate predictions provided by the GNN-SAGE model show potential for real-time water quality management, aiding in developing regulatory guidelines for adaptive road salt management plans to better protect vulnerable aquatic freshwater ecosystems in urban streams from extreme events.

**Author Contributions:** Conceptualization, J.V.G.T. and B.G.; methodology, P.A.C.R., J.V.G.T. and B.G.; software, P.A.C.R.; validation, P.A.C.R., J.V.G.T. and B.G.; formal analysis, P.A.C.R.; investigation, P.A.C.R., J.V.G.T. and B.G.; resources, J.V.G.T. and B.G.; data curation, J.V.G.T. and B.G.; writing—original draft preparation, V.O.S. and P.A.C.R.; writing—review and editing, V.O.S., P.A.C.R., J.V.G.T. and B.G.; visualization, V.O.S. and P.A.C.R.; supervision, J.V.G.T. and B.G.; project administration, J.V.G.T. and B.G.; funding acquisition, B.G. and J.V.G.T. All authors have read and agreed to the published version of the manuscript.

**Funding:** This research study was funded by the Natural Sciences and Engineering Research Council of Canada (NSERC) Alliance, grant No. 401643, in association with Lakes Environmental Software Inc., and by the Conselho Nacional de Desenvolvimento Científico e Tecnológico—Brasil (CNPq), grant no. 303585/2022-6.

**Data Availability Statement:** The original dataset can be retrieved from https://cvc.ca/real-time-monitoring/ (accessed on 26 July 2023). The algorithms and datasets used can be downloaded from https://drive.google.com/drive/folders/136RH-G-nPVScO7Ln7OOC0WEYl3kk5kDW and https://drive.google.com/drive/folders/13Ef-_EklzJze8pZx1oIDoQFKU304d7NF, respectively (accessed on 28 August 2023).

**Conflicts of Interest:** The authors declare no conflict of interest.

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
