# Peer review of "Graph-Based Deep Learning Model for Forecasting Chloride Concentration in Urban Streams to Protect Salt-Vulnerable Areas"

_environments, doi:10.3390/environments10090157_

Round 1
Reviewer 1 Report
The objective of this study was to use deep learning approaches for forecasting the chloride concentrations in an Urban stream in Canada. The study is interesting and technically sound. However, there are a few issues that need to be addressed to improve the quality of the study.
1. Introduction:
I am missing the novelty of this study here. As the authors mentioned, there have been past studies that used machine learning methods to predict chloride concentrations. I found another study (Bilali & Taleb, 2020) where the authors used 8 machine learning models to predict 10 water quality parameters, including chloride. Is the use of graph theory a novelty of this study? The authors need to highlight how their approach is unique or different from past studies.
El Bilali, A., & Taleb, A. (2020). Prediction of irrigation water quality parameters using machine learning models in a semi-arid environment. Journal of the Saudi Society of Agricultural Sciences, 19(7), 439-451.
2. Materials and Method:
Page 3, lines 114-115: How frequently was the data collected? Please add more details like the type of water physical-chemical parameters collected, analysis method, and the frequency of collection. For example, how often the chloride data was collected and what method was used to collect and analyze the data.
Page 3, lines 133-136: how many data points were used for the training and validation?
Please add a flow chart outlining the various tasks performed during this study. This might clarify the methodology better.
3. The figure quality needs to be improved.
a) Figure 2: The resolution needs to be improved to read the numbers and letters clearly. You can also increase the font and picture size for better visualization.
b) Figure 3: axes titles are too small to read. Please increase the font. Please remove the background shedding for better visualization.
c) Figures 9, 11, 12, and 13: axes titles are too small to read. Please increase the font size.
I also sugguest that the manuscript to be edited by a professional editor or a native English speaker since there are several grammatical errors and awkward sentence structures (making it hard to understand) on the manuscript.
Author Response
Dear reviewer,
Thank you very much for your thoughtful comments. Please, see the attachment for our reply to your suggestions.
Best wishes,
The Authors.

Reviewer 2 Report
The authors present novel deep learning approaches for forecasting Cl concentrations in urban streams. This is an important area of research, and the technical work was well done. I have a few minor issues with the discussion but overall the paper is in good shape. Specific issues:
1. The language was a bit terse for water quality modelling experts who are not experts in machine learning. Can you present error metrics typically used by other water quality modellers alongside the metrics you do present, e.g. Nash-Sutcliffe and/or Kling-Gupta Efficiency? This will enable a better comparison with non-ML models of Cl. Also, with your Table 1, can you include results from INCA-Cl, a process model of Cl dynamics, for comparison purposes?
2. The use of lags and the data were a little hard to follow. Can you include a table detailing all the variables used in the models, i.e. something like the figure presenting the SHAP analysis? I did not realize till then that you used Cl lagged a number of times, and did not know which stations you used the data from. Also, please number the stations on the map.
3. I think the discussion is the only part of the paper that needs any substantive work. It is pretty thin. I was left wondering a few things:
3a how does the model performance of the ML models compare to other process models?
3b do the authors think of this as a complex time series model, kind of like an auto-regression but not linear? It seems to me to be similar to that. It begs the question of how long in the future this model could project - 12 hrs is not long.
3c what kinds of applications do the authors envision with this model? I am not sure what can be done with it.
3d what do the authors think about the model's ability to predict toxic exceedance events? This is the management metric most of interest (e.g. Cl exceeding the chronic and acute levels of I believe 120 mg/l and 640 mg/l).
Those were my main questions. A more fulsome discussion is all the paper really needs to be publishable. The rest of the paper is already very high quality, and would benefit from the minor improvements I suggested.
Author Response

(The authors gave the same response as above.)

Reviewer 3 Report
This manuscript, environments-2566489-peer-review-v1- entitled "Graph-based deep learning model for forecasting chloride concentration in urban streams to protect salt-vulnerable areas," is well written and has potential, but it should be more organized. This research investigates the the proposed GNN-SAGE is compared to other models, including a Deep Neural Network based transformer (DNN-Transformer) and a benchmarking persistence model for 6 hours forecasting horizon
In my opinion, a careful revision of the English language should be carried out as there currently are some unclear sentences. The study seems to be well designed. The methodology and results are technically sound. Discussions on the scientific and practical values of the study, the limitations of proposed models, and future work are meaningful. I recommend accepting this manuscript after revision. The main concerns are as follows:
1) Quantitative results should be provided in the abstract to make it more comprehensive. Results of DM models Should be added in the abstract section. Also, The main aim of the study should be clearly mentioned in the abstract.
2) More literature review about the other methods is needed. The manuscript could be substantially improved by relying and citing more on recent literature about contemporary real-life case studies of sustainability and/or uncertainty, such as the followings.
· Samani, S., Vadiati, M., Nejatijahromi, Z., Etebari, B., & Kisi, O. (2023). Groundwater level response identification by hybrid wavelet–machine learning conjunction models using meteorological data. Environmental Science and Pollution Research, 30(9), 22863-22884.
3) Credit River is adopted as the case study. What are other feasible alternatives? What are the advantages of adopting this case study over others in this case? How will this affect the results? The authors should provide more details on this.
4) Please provide all software and packages used in this study.
5) It is important to give a better description of the samples and the sampling protocol since we are trying to understand the data variability. What are the advantages of adopting these parameters over others in this case? How will this affect the results? More details should be furnished.
6) It is better to add more error criteria such as BIC or AIC to better understand the model's ability.
7) The discussion section in the present form is relatively weak and should be strengthened with more details and justifications.
8) The limitations of the present study should be added to the paper, specifically for further research.
Author Response

(The authors gave the same response as above.)
